# The Role of Urinary VEGF in Observational Studies of BPS/IC Patients: A Systematic Review

**DOI:** 10.3390/diagnostics12051037

**Published:** 2022-04-20

**Authors:** Pedro Abreu-Mendes, Aurora Costa, Ana Charrua, Rui Almeida Pinto, Francisco Cruz

**Affiliations:** 1Department of Urology, São João Universitary Hospital Center, 4200-319 Porto, Portugal; ruipinto@mac.com (R.A.P.); cruzfjmr@med.up.pt (F.C.); 2Faculty of Medicine, University of Porto, 4099-002 Porto, Portugal; aurorambgc@gmail.com; 3Translational Neurourology Department of I3S, University of Porto, 4099-002 Porto, Portugal; anacharr@gmail.com; 4Instituto de Biologia Molecular e Celular, University of Porto, 4099-002 Porto, Portugal

**Keywords:** interstitial cystitis, bladder pain syndrome, urinary VEGF, VEGF receptors

## Abstract

Background: Bladder pain syndrome/interstitial cystitis (BPS/IC) is a chronic pain condition, often underdiagnosed, with an important impact on patient quality of life. More recently, an association between VEGF and its receptors has been suggested in BPS/IC pathophysiology, due to their role in promoting angiogenesis and inflammation, which can enhance bladder pain. Eventually, VEGF may be used as a biomarker for the diagnosis and prognostication of BPS/IC. To further clarify this issue, this review aims to critically summarize the available information, giving rise to a solid starting point for future studies. Methods: We systematically searched PubMed and Embase, using the queries “urinary VEGF”, “urinary VEGF” AND “pain”, “urinary VEGF” AND “lower urinary tract symptoms” and “urinary VEGF” AND “LUTS” from January 2016 to February 2022. Results: A total of 1026 papers were identified from which 7 articles were included in this study, which assessed 1036 participants. Regarding VEGF levels, overactive bladder (OAB) and healthy patients were used for comparison with BPS/IC patients. VEGF concentration seems to be higher when compared to healthy patients and overactive bladder (OAB) patients. Higher levels of VEGF were associated with pain severity, while a decrease in VEGF concentration was associated with pain and symptom improvement in women. However, these findings were not constant in all studies. Conclusions: There is a trend toward a relevant association between increased VEGF levels and pain or symptom severity in BPS/IC patients. Although there are some discrepancies among the studies and the number of patients included is small, VEGF and its receptors should be considered for future studies regarding its use in BPS/IC pathophysiology, diagnosis and prognostication.

## 1. Introduction

Bladder pain syndrome/interstitial cystitis (BPS/IC) is a chronic pain condition, often underdiagnosed, that affects primarily women [1]. It is a symptomatic-based syndrome in which the main complaint is bladder-related pain or discomfort, accompanied by other storage lower urinary tract symptoms such as polyuria and nocturia [2]. The pain is usually relieved after emptying the bladder. Since BPS/IC is an exclusion diagnostic [3], it can only be performed after ruling out other causes such as infection, cancer and gynecological conditions, among other pelvic and neurological painful conditions [4,5,6]. This entity has an important impact on the quality of life of these patients, since it may lead to a decrease in work productivity, sleep quality and sexual dysfunction, among other factors [7]. Considering the current evidence, this condition is phenotyped in two subgroups that are defined by the presence or absence of Hunner’s lesions [8]. In the classic definition provided by the European Society for Study of Interstitial Cystitis (ESSIC), the cystoscopy findings are divided into three groups, and only when Hunner’s lesions are present, it could be defined as IC, and the category in these criteria is 3X. The other two types of findings, 1X and 2X, correspond to normal urothelium or glomerulations, respectively, and define the disease as BPS [9,10]. Of note, in some guidelines, patients with normal cystoscopy findings and normal biopsies (ESSIC 1A) are considered to have “hypersensitive bladder”, as it will appear later in the paper [3]. The Hunner’s lesions are characterized by a reddened concentric vascularized area that upon distension of the bladder cause urothelium split and bleeding [11]. Bleeding can be explained by angiogenesis that leads to glomerulation (factor required for the diagnosis of IC), which is evidence of blood vessel fragility [12]. Moreover, this condition is associated with urothelial hyperpermeability, allowing small ions passage to the bladder wall where they can activate nociceptors already sensitized by inflammatory agents [13]. As a matter of fact, inflammation in BPS/IC patients is associated with an increased tissue concentration of inflammatory cytokines [14,15].

Previous studies showed that bladder changes could have the participation of vascular endothelial growth factor (VEGF) and its receptors (VEGF-R) [16]. The presence of VEGF may contribute to hyperalgesia experienced by the patients. VEGF seems to be associated not only with the vascular genesis but also with neuronal control and nerve repair [17,18]. It has been also reported that there is an up-regulation of HIF-1α (Hypoxia-inducible factor 1-alpha) and, consequently, overexpression of VEGF, since HIF-1α acts as a transcriptional mediator to VEGF in the presence of hypoxia. This may be more relevant in umbrella apical cells where hypoxia may occur, related to lower bladder perfusion during the filling phase [19].

A previous study demonstrated that the intravesical instillation of VEGF in the bladder could cause changes similar to Bacillus Calmette−Guerin bladder instillations, with reduced time between micturitions and lower voiding pressures [20]. Additionally, an up-regulation of sodium channels at the level of bladder dorsal root ganglia neurons was noted, associated with augmented abdominal sensitivity to mechanical stimulation [20].

Besides the VEGF molecule and its isoforms, a total of three membrane VEGF receptors were identified [21]. While the role of VEGFR1 in angiogenesis is uncertain, other VEGF receptors such as VEGFR2 and VEGFR3 were proven to have an important part in angiogenesis [22,23]. The VEGFR1 is expressed in sensory neurons and monocyte/macrophages, promoting inflammation and pain, although some evidence seems to point toward this receptor being a “decoy”, blocking the VEGF ligation to other subtypes related to angiogenesis [17]. The second subtype is the main one responsible for promoting angiogenesis and increasing vascular permeability, and it is produced in endothelial cells. Finally, VEGFR3 is responsible for generating the main signal for lymphangiogenesis [24]. Both VEGFR1 and VEGFR2 may be found in urothelial cells. More importantly, there is a soluble isoform of the VEGFR1 (sVEGF-R1), which is responsible for trapping VEGF, preventing it from activating the membrane receptors [21,25]. These facts support that this pathway, involving VEGF and its receptors, could be relevant as a possible target in the management of this disease [24].

Considering everything mentioned above concerning urinary VEGF and receptors and its potential use in BPS/IC disease management (a subject on which there is a lack of recent studies), this review aims at critically summarizing the available information, giving rise to a solid starting point for future studies.

More precisely, we focused our search on patients diagnosed with bladder pain syndrome, comparing them to healthy patients or patients with other urinary conditions reported, regarding urinary VEGF and its receptors levels. We anticipate the existence of clinical evidence on a statistically significant difference in urinary VEGF and its receptors levels between the compared groups.

## 2. Materials and Methods

This study took into account the preferred items for systematic reviews and meta-analyses (PRISMA) statements [26].

Until 28 February 2022, we systematically searched PubMed and Embase, using the queries “urinary VEGF”, “urinary VEGF” AND “pain”, “urinary VEGF” AND “lower urinary tract symptoms” and “urinary VEGF” AND “LUTS” (no Mesh Terms were available), regarding English articles and the time frame of January 2016 to February 2022. Once the searches were combined, titles and abstracts were retained. In the identification phase, a total of 1026 papers were obtained. Then each investigator independently proceeded to the screening phase as a way of minimizing selections influenced by a single-person bias [27]. Every duplicate and every study that was not related to pain (e.g., related to diabetes, hypertension or bladder cancer) or that was in the pre-clinical stage was excluded, for a total of 1014 papers (393 for the pre-clinical stage, 299 for non-pain related studies, 322 for duplicates). Twelve (12) eligible articles remained. The inclusion criteria were aimed at studies whose results were published not only as full articles but as abstracts as well; given the low number of publications in this area, the authors preferred to use all the possible information. In fact, studies in which full articles were not available were considered for further analysis as well, knowing the impossibility to correctly evaluate them. We considered studies that involved patients diagnosed with bladder pain syndrome without any restriction at the moment of the diagnosis. The comparing group could be healthy patients or patients with other urinary conditions. The objective was to search for studies regarding urinary VEGF and, when present, its receptor level changes between groups, since it was expected that there could be a significant difference between them. Due to the lack of studies in this specific subject (especially in the clinical stage), only studies that followed case-control, cohort or cross-sectional study design were found and analyzed. Finally, we had a total of 7 included articles for thorough analysis: six full articles and one abstract for which the integral article was not accessible for complete analysis. Those articles were, once again, analyzed independently by the two investigators (P.A.M and A.C.) as a way of minimizing bias induced by a single-person analysis [27]. Each of them collected data regarding the studies’ characteristics such as title, first author name, publication year, study design, quality assessment (using quality assessment tools later mentioned), sample size and characteristics, and study aim. They also collected important findings and limitations for each study analyzed. If there was a discrepancy between the two investigators’ collected information, a third one would intervene.

When analyzing the studies, once more, we confirmed whether it involved patients with bladder pain diagnosis. The articles which subdivided these patients into different groups, such as patients with Hunner’s lesions (IC) and without Hunner’s lesions (BPS), were considered. We simplified and considered them both as patients with BPS/IC. The idea was to measure urinary VEGF and its receptor levels to verify if there were any changes in them as an outcome when comparing different groups. Once more, BPS/IC patients could be compared to healthy patients or patients diagnosed with another pathology.

For quality and risk of bias assessment for each type of study, we used previously published checklists. More specifically, for case-control and cohort studies, we used the Newcastle Ottawa Scale (NOS) [28], since it is the most known scale for assessing bias and quality in observational studies; it is easy to use and can be applied to cohort and case-control studies [29]. While reading the papers, we were able to confirm whether the item was met in the article. Regarding cross-sectional studies, we used JBI’S checklist [30]. These quality assessments are stated in Appendix A (Table A1, Table A2 and Table A3, respectively).

When considering the abstract also included in this review, several of the quality assessment parameters were not available for analysis. However, considering its relevance for the present review, we chose to include it.

## 3. Results

### 3.1. Study Selection

A total of 1026 papers were obtained (652 from PubMed and 374 from Embase). Of these, we proceeded to the screening in which we excluded duplicates (*n* = 322) and any study regarding non-pain-related issues (*n* = 299; e.g., issues related to diabetes, bladder cancer and hypertension) or in the pre-clinical stage (*n* = 393; involving in vitro and animal studies). In the next phase, we had 12 eligible articles. Upon the full reading of the papers taken into account, we considered 7 to be included in this systematic review: six (6) full articles [31,32,33,34,35,36] and one (1) abstract [37] of which the integral article was not accessible for complete analysis. The article selection process is stated in the flow diagram shown in Figure 1.

The articles included in this review were submitted to a quality assessment as already stated (see Table A1, Table A2 and Table A3 from Appendix A).

### 3.2. Studies’ Overall Characteristics

As for study characteristics, the data regarding title, first author, publication year, sample size, sample characteristics, study aim and quality assessment were also collected and gathered in a table for easier access (see Table A4). We included mostly papers with good quality (we were not able to check for quality parameters for only one of the studies [37]). Moreover, five papers followed a case-control design [31,32,33,34,37], while one was a cohort study [35], and the other one left was a cross-sectional one [36]. One of the studies involved a big sample size (*n* = 491) [31]. However, the others had smaller sample sizes (maximum *n* = 216 [35] and minimum *n* = 36 [33]). A total of 1036 participants were involved. Each study aim is also reported in Table A4 (Appendix B).

Caution must be taken in the nomenclature used. Some studies use the term Urological Chronic Pelvic Pain Syndrome (UCPPS) to encompass both males with chronic prostatitis/chronic pelvic pain syndrome (CP/CPPS) and female patients with BPS/IC. Furthermore, in some studies, IC patients are called HIC, and BPS patients are named as NHIC. The last nomenclature (HIC and NHIC) is not used in the manuscript.

### 3.3. Individual Characteristics, Findings and Limitations

The information regarding each study’s characteristics is shown in Table A4. As for the relevant findings regarding urinary VEGF and its receptors, other findings (such as questionnaires provided during the study) and limitations can be stated in Table A5 (Appendix B).

The first study analyzed was “Identification of novel non-invasive biomarkers of urinary chronic pelvic pain syndrome: findings from the Multidisciplinary Approach to the Study of Chronic Pelvic Pain (MAPP) Research Network”, a case-control study by Dagher et al. published in 2017 [31]. We classified it as having a quality of 8/9. This article addresses the identification of urinary biomarkers, such as VEGF and VEGF-R1, NGAL (Neutrophil gelatinase-associated lipocalin), and MMP-9/NGAL (Matrix metallopeptidase 9/Neutrophil gelatinase-associated lipocalin) complex in UCPPS, including both female and male patients. Here, they considered in three groups (UCPPS patients for patients who reported a non-zero score on pelvic pain scale; Positive Controls defined for the presence of non-urological associated syndrome; Healthy Controls for patients without any pathology), with a total of 491 patients. A clean-catch midstream urine was collected. The authors reported that although urinary VEGF concentration in females was significantly higher in UCPPS patients than in HC (mean UCPPS = 3.72, mean HC = 3.46; *p* = 0.0076), it did not distinguish female UCPPS from HC overall (AUC = 0.58). Males with UCPPS reported higher VEGF (mean UCPPS = 1.78, mean HC = 0.98, *p* < 0.0001) and VEGF-R1 (mean UCPPS = −0.33, mean HC = −1.25, *p* = 0.004) levels compared to healthy controls. Moreover, in univariable analysis, significance for a relation between increased urinary normalized VEGF concentration and pain severity in women was found (β = 1.280, *p* = 0.0127). No association was found after adjustment for other biomarkers (*p* = 0.2502). As a limitation in this study, only UCPPS with more severe urologic pain were included to maximize the potential to find significance.

“Comparison of inflammatory urine markers in patients with interstitial cystitis and overactive bladder” is a case-control study published in 2018 by Furuta et al. (with a quality rating of 7/9) [32]. In this paper, the authors analyzed the results based on the following conditions: 30 patients with BPS, 30 patients with IC and 28 patients with OAB syndrome. In this case, they obtained the levels of 40 urinary biomarkers. Mid-stream urine was collected. Two questionnaires were handed out to patients: the O’Leary and Sant Symptom and Problem Index (divided into two parts, OSSI and OSPI) and the Visual Analogue Scale (VAS). Both assessed pain and bothersome levels associated with BPS/IC. In all the questionnaires, higher score values were indicative of more severe symptoms: VAS from 0 to 10, OSPI from 0 to 16 and OSSI from 0 to 20. Regarding urinary VEGF levels, they were significantly increased in both IC (mean 48.7 ± 28.1) and BPS patients (mean 69.7 ± 40.1). When compared to urinary levels of VEGF in OAB patients (mean 19.5 ± 7.5), both showed a statistically significant difference (*p* < 0.05), suggesting that BPS/IC patients present a severe chronic inflammatory bladder state. For that reason, an increased level of VEGF was considered BPS/IC-specific. There was no significant difference between the IC and BPS groups. Regarding the scores of the questionnaires, when compared to the OAB group (OSSI-1.1 ± 1.1; OSPI: 0.6 ± 1.1; VAS: 0.4 ± 0.9), both groups (IC and BPS) had higher scores for OSSI and OSPI, while also reporting higher VAS scores for pain. More precisely, the results were IC (OSSI: 10.3 ± 5.5; OSPI: 9.0 ± 4.8; VAS: 5.5 ± 3.3) and BPS (OSSI: 8.1 ± 4.4; OSPI: 7.6 ± 4.9; VAS: 4.8 ± 3.3), with all the parameters showing a statistically significant difference (*p* < 0.01) when compared to OAB patients. As a limitation, this study was performed in a small sample and, since they were already diagnosed, they had already received some treatment for the respective disease, which could also have influenced the results. These facts were not considered when the statistical analysis was performed. Furthermore, previous episodes of hydrodistension could have interfered with the results, even though investigators tried to minimize this limitation by waiting 3 months after the last hydrodistension episode before collecting the samples and the data.

As for “Angiogenesis in bladder tissues is strongly correlated with urinary frequency and bladder pain in patients with interstitial cystitis/bladder pain syndrome” by Furuta et al., it is a case-control study, published in 2019, classified as 6/9 [33]. The study aimed to determine whether there was an association between bladder inflammation, angiogenesis, fibrosis, and urothelial denudation in biopsied bladder specimens when comparing BPS/IC patients with controls. VEGF was measured directly from bladder specimens. The authors considered three groups with twelve female patients each: one with BPS patients, the second with IC and the third one with healthy controls. Bladder specimens were collected from each patient and underwent an immunohistochemical analysis for the expression of VEGF; other biomarkers, such as TNF-α (Tumor necrosis factor alfa), mast cell tryptase, CD31 (Cluster of differentiation 31), TGF-β (Transforming growth factor-beta) and E-cadherin, were also assessed. Then, two questionnaires regarding IC symptoms and problems were given to the patients (OSSI and OSPI, respectively) while the VAS (Visual Analogue Scale) score was used for assessing bladder pain. As for the results, VEGF levels were statistically significantly higher in IC and BPS patients when compared to the control group (*p* < 0.01 for both cases). Additionally, BPS and IC patients had higher OSSI, OSPI and VAS mean scores. More precisely, the results for IC were (OSSI: 15.9 ± 2.6; OSPI: 15.0 ± 1.2; VAS: 9.4 ± 0.9) and for BPS were (OSSI: 9.3 ± 4.7; OSPI: 8.5 ± 4.7; VAS: 5.9 ± 2.1), all of them with a significance of *p* < 0.01 when compared to the respective questionnaires of the control group. Therefore, these results lead to new insights on basic pathophysiology in BPS/IC. As the main limitation, this study presented a small sample size for each group.

Regarding “Molecular Taxonomy of Interstitial Cystitis/Bladder Pain Syndrome Based on Whole Transcriptome Profiling by Next-Generation RNA Sequencing of Bladder Mucosal Biopsies”, it is a case-control study, published in 2019 by Akiyama et al. (with 7/9 in quality assessment) [34]. In this study, a total of 42 women were assessed and divided into four groups: IC, BPS (in this study, it was needed for the patient to have inflammation in bladder biopsies to be considered BPS), hypersensitive bladder (HSB) (a term usually used in Asian guidelines for BPS patients with normal cystoscopy and normal bladder biopsy) and controls. The groups enrolled 12, 11, 10 and 9 patients, respectively. Bladder mucosae were biopsied from consecutive patients. As for VEGF relative expression, the following results were found: IC (Median 0.412; 99% CI 0.035–0.048), BPS (Median 0.014; 99% CI 0.013–0.018), HSB (Median 0.016; 99% CI 0.012–0.024) and controls (Median 0.014; 99% CI 0.014–0.018). The authors report that the expression of VEGF was higher in IC patients when compared to BPS (*p* = 0.002), HSB (*p* = 0.004) and controls (*p* = 0.007). There was no significant difference when comparison was carried out between BPS, hypersensitive bladder and controls. Another statistical analysis was performed with the intent to verify whether there was a relation between VEGF levels in IC patients and symptom severity. Higher VEGF expression was significantly associated with symptom severity, revealing a positive correlation with storage symptoms (ρ = 0.47; *p* < 0.01), voiding symptoms (ρ = 0.40; *p* = 0.02), daytime frequency (ρ = 0.35; *p* = 0.03) and nocturia (ρ = 0.40; *p* = 0.01). Lower bladder capacity was correlated with higher VEGF urinary levels (ρ = −0.46; *p* < 0.01). The authors also stated a significant association between higher VEGF levels and higher OSSI (ρ = 0.44; *p* < 0.01), OSPI (ρ = 0.46; *p* < 0.01) and VAS pain (ρ = 0.49; *p* = 0.001) scores, indicating a positive correlation. To evaluate if there were significant differences between the groups in the questionnaires scores, the authors performed a pairwise comparison between IC, BPS, HSB and control groups. The results for each questionnaire were OSSI (IC: 15.8 ± 3.7; BPS: 12.3 ± 3.0; HSB: 9.8 ± 4.8; control: 3.3 ± 1.4), OSPI (IC: 13.3 ± 2.1; BPS: 12.4 ± 2.3; HSB: 10.7 ± 2.6; control: 1.8 ± 2.0) and VAS pain score (IC: 8.2 ± 2.3; BPS: 5.7 ± 3.3; HSB: 4.9 ± 2.4; control: 0); they all had a *p* < 0.01 significance. These results point out a significant difference in each questionnaire assessment between all four groups, with the IC group presenting the highest scores. The relation between microvessel density and VEGF expression was also assessed, revealing a positive and significant association between them (*p* < 0.0001). There were some limitations in this study, such as the small sample involved and the fact that some samples involved an admixture of urothelium and lamina propria, which could mean that there was an underestimation of the results regarding markers expression due to inappropriate samples.

The fifth paper included was “Association of Longitudinal Changes in Symptoms and Urinary Biomarkers in Patients with Urological Chronic Pelvic Pain Syndrome: A MAPP Research Network Study”, which was published in 2021; it is a prospective cohort study by Roy et al. [35] with a quality rating of 7/9. The authors randomly selected 126 UCPPS patients from a previously defined cohort: 116 were female and 100 were male patients [38]. In this study, the aim was to analyze a series of non-invasive biomarkers for their ability to objectively monitor the longitudinal clinical status of UCPPS patients. A urine sample was collected non-invasive way from a group of 216 UCPPS patients at three time points: at baseline (0), 6 months later and then at the end of 12 months. The samples were analyzed in duplicate and in a double-blinded manner as a way of minimizing bias. An odds ratio (OR) > 1 meant a greater likelihood of symptom improvement with an increase in a certain protein, whereas an odds ratio (OR) < 1 meant a greater likelihood of symptom improvement with a decrease in its concentration. It was shown that most regression or progression occurs within the first 6 months. More specifically, the results indicate that a decrease in the urinary VEGF was associated with a greater likelihood of urologic pain improvement (OR = 0.78; *p* = 0.03). On the other hand, there was not a significant association between VEGF-R1 and overall urologic pain symptoms (OR = 0.88; *p* = 0.06). For every finding mentioned above, it was proposed in this study that the measurement of both urinary VEGF and VEGF-R1 (in a non-invasive and inexpensive way) could be used to monitor progression or treatment efficacy. Besides, since it may play an important role in UCPPS pathophysiology, it was suggested as a possible target for future treatments.

A possible limitation for this study is suboptimal reliability in data concerning some of the biomarkers, such as MMP biomarkers.

Finally, as for the article “Relationship of Bladder Pain With Clinical and Urinary Markers of Neuroinflammation in Women With Urinary Urgency Without Urinary Incontinence”, a cross-sectional study by A. Soriano et al., published in 2021 [36], it involved 101 women (with age superior to 18 years) with mild urinary urgency without incontinence [38]; 62 were considered to present bladder pain. The aim was to assess if bladder pain is associated with the presence of neurogenic inflammation in the bladder wall and neuroinflammatory biomarkers in the urine in women with urinary urgency without incontinence. They submitted one questionnaire for the determination of urinary symptoms (F-GUPI), one for the determination of interstitial cystitis problems (OSPI), one for assessment of the interstitial cystitis symptoms (OSSI) and one questionnaire concerning neuropathic pain (PainDETECT). Midstream urine was collected. They also provided a sample of urine for urinary neuropeptides, including VEGF and NGF (nerve growth factor), BDNF (brain-derived neurotrophic factor) and Opn (urinary osteopontin) measurement. Only 84 patients provided urine samples. After applying a Visual Analogue Scale for pain, the women included in this study were divided into two groups (BP—bladder pain; NBP—non-bladder pain); those groups were later compared. As for the results, the group classified as BP reported higher scores (and therefore worse symptomatology and quality of life) in all the questionnaires. More specifically, the reported results were (mean BP = 28.1, mean NBP = 12.3, *p* < 0.001) for F-GUPI, (mean BP = 9.8, mean NBP = 6.7, *p* < 0.001) for OSSI, (mean BP = 8.7, mean NBP = 4.9, *p* < 0.001) for OSPI and (mean BP = 13.3, mean NBP = 5.1, *p* < 0.001) for PainDETECT. Considering the measurement of urinary biomarkers, elevated normalized VEGF levels were significantly associated with bladder pain (β = 0.04 pg/mg Cr, 95% CI 0.001–0.07, *p* = 0.04). In this study, urinary VEGF measurements were considered as a possibility for phenotyping or a way of following patients’ responses to treatments. Possible confounders were taken into account while performing statistical analysis. However, as a limitation, no healthy control group was present, and the total cohort was small.

Even though we were not able to access the complete article “Relationship of Pain Catastrophizing With Urinary Biomarkers in Women With Bladder Pain Syndrome” (just the abstract), we decided to include this abstract due to its significant relevance in the subject approached in this review [37]. This study, written by Soriano et al., followed a case-control study design and was published in 2021. No quality assessment was possible, since there was no access to the full article. However, the results indicate a higher pain catastrophizing score associated with lower VEGF levels (*p* = 0.03), while bladder pain was associated with higher VEGF levels (*p* = 0.01). Moreover, higher catastrophizing scores were significantly associated with worse urinary symptoms, greater pelvic and neuropathic pain and worse quality of life scores (all *p* < 0.01). There was no possibility to assess this study’s specific limitations since we did not access the rest of the article.

Overall, in all the studies analyzed, VEGF and VEGF-R1 (in some studies) had significantly elevated or higher levels (if compared to healthy controls or patients diagnosed with other pathologies) in BPS/IC patients. Furthermore, it was stated that a decrease in urinary VEGF concentration could lead to symptom improvement in women. On the other hand, in some of the studies, we were able to see a positive association between BPS/IC groups and OSSI, OSPI and VAS scores. Finally, lower levels of VEGF were significantly associated with higher pain catastrophizing personality. However, it is important to remember the impossibility to access the full article regarding the later results. The small samples used in most of the studies are a limitation across most studies.

## 4. Discussion

There is a lack of understanding of BPS/IC pathophysiology. As previously stated, it encompasses bladder-related pain or discomfort, and it is frequently accompanied by other storage lower urinary tract symptoms [2,4]. Even though it affects primarily women, and contrary to previous beliefs, pain severity was reported to be similar in both genders, while bladder-focused symptoms were more frequently reported by women [39]. On the other hand, there is an increasing disease approach based on biomarkers, as they could be truly valuable in a condition with a symptom-based diagnosis, that is, lacking robust phenotyping and having multiple confounding diseases. According to our review, VEGF and its receptors may have an important pathophysiological role. As reported, high levels of VEGF are associated with pain severity, while a decrease in VEGF concentration is associated with pain and symptom improvement in women [31]. In addition, its concentration seems to be higher when compared to healthy patients [33] and OAB patients [36]. An association between increased VEGF levels and higher OSS scores and VAS pain scores was also seen. Since VEGF urinary levels would allow a frequent and economical assessment, its use in BPS/IC diagnostics, relapse prediction and management should be considered in the future. As previously stated, VEGF is associated with angiogenesis [22], and it is proven that its inhibition leads to a decrease in inflammation [40]. Therefore, it would be potentially beneficial to consider anti-angiogenic therapy as a method of disease management. This strong VEGF expression in IC patients could be caused by the hypoxia in bladder tissues during the filling phase, leading to upregulation of urinary VEGF expression, causing bladder fibrosis and a decrease in capacity.

All the papers present in this review took into account the association between increased levels of VEGF and BPS/IC symptom severity, while only two showed an association between increased levels of VEGF-R1 [31,35] and BPS/IC symptoms. Indeed, it is important to consider its role in pain regulation, since VEGF-R1 contributes to VEGF-R2 response modulation, thereby influencing pain [17]. Moreover, as previously reported, intravesical instillation of VEGF leads to an increase in bladder sensory fiber density and the development of pelvic hypersensitivity in mice. Furthermore, its inhibition through a systemic treatment with VEGF-neutralizing antibodies in an animal model reduced pelvic hypersensitivity [41].

It is important, however, to highlight the lack of studies regarding urinary VEGF receptors. One of the two studies here presented had some limitations regarding the cases’ representativeness and data reliability [31]. For that reason, further studies involving bigger sample sizes would be necessary to verify the results here presented.

In this review, we faced some limitations. Indeed, some of the studies comprised a small sample [32,33,34,36]. Overall, even though we were only able to include seven articles, the majority (6) were considered to have good quality when the quality assessment was performed. In one case, we could only have access to the article’s abstract, missing potentially important information.

When there was a subdivision of patients between IC and BPS, we opted for a simplification by consider them all as BPS/IC patients when discussing the results. However, the fact that these two groups could potentially represent different entities should be mentioned [42] since some studies reported a tendency to present pancystitis, frequent clonal B-cell expansion and epithelial denudation [43] in patients with Hunner’s lesions present. For that reason, it would be beneficial to keep that in mind when creating future studies regarding this subject.

## 5. Conclusions

There was a tendency toward a significant association between increased VEGF levels and pain or symptom severity, even though it was not equal in every study. Although there is a small amount of data here analyzed regarding this subject among the results presented, VEGF and its receptors should still be considered for future studies regarding its use in BPS/IC diagnostics and management. However, these studies must involve bigger samples and follow a study design with better quality of clinical evidence as a way of minimizing limitations.

## Figures and Tables

**Figure 1 diagnostics-12-01037-f001:**
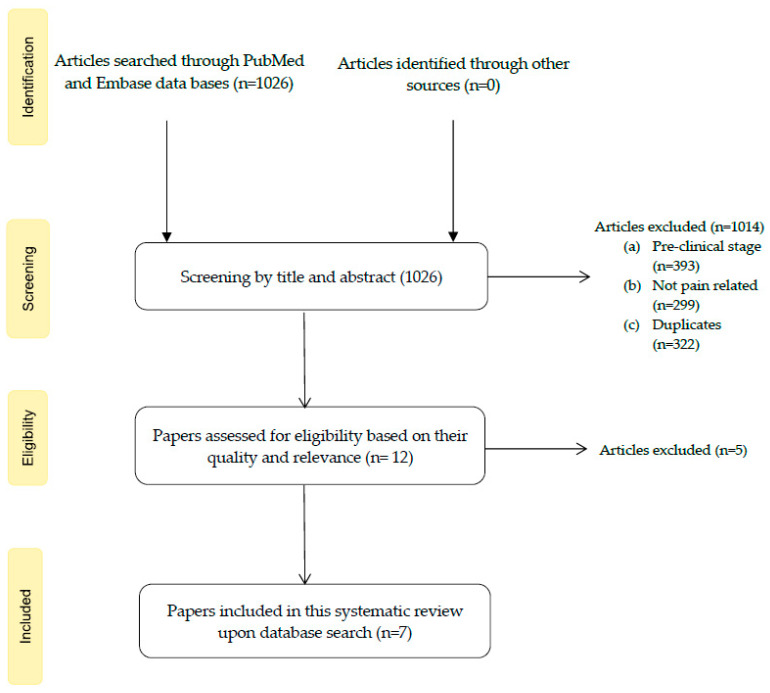
Flow diagram showing the study selection process.

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
