# Peer review of "The Role of Urinary VEGF in Observational Studies of BPS/IC Patients: A Systematic Review"

_diagnostics, 2022, doi:10.3390/diagnostics12051037_

Round 1

Reviewer 1 Report

  1. Please review the definition of Hunner lesion (Classic definition). In addition, the term 'Hunner's lesion' and 'Hunner lesion' are heterogeneously used in the manuscript.
  2. If possible, access to the manuscript (which is included in the study as abstract) is recommended.
  3. How many investigators reviewed the screened papers? Most systematic reviews present the information of the reviewer in initials. Why was this study not registered in the registry? 
  4. The problem with interstitial cystitis/bladder pain syndrome and other pelvic pain disorders are overlapping symptoms and difficulty in differential diagnosis. The largest study which included UCPPS patients included CP/CPPS patiens which might include subjects who are not exactly  IC/BPS population. Please provide the definition of diagnosis in each study. It is very difficult to read and figure out such information in current manuscript. 
  5. The site where VEGFs were evaluated varies among studies - urine & bladder. Please also provide the methodology of acquiring the specimen in each study.
  6. Overall, this study seems to be a review article rather than systematic review.

Author Response

  1. Please review the definition of Hunner lesion (Classic definition). In addition, the term 'Hunner's lesion' and 'Hunner lesion' are heterogeneously used in the manuscript. - Both done
  2. If possible, access to the manuscript (which is included in the study as abstract) is recommended - The abstract is provided online doi:10.1097/SPV.0000000000001041. We tried to contact the author and the co-authors but they didn't answer, unfortunately. 
  3. How many investigators reviewed the screened papers? Most systematic reviews present the information of the reviewer in initials. Why was this study not registered in the registry? They were analyzed by me (First author) and Dr. Aurora Costa - We added this information.The study was not registered given the lack of time for the publication.
  4. The problem with interstitial cystitis/bladder pain syndrome and other pelvic pain disorders are overlapping symptoms and difficulty in differential diagnosis. The largest study which included UCPPS patients included CP/CPPS patiens which might include subjects who are not exactly  IC/BPS population. Please provide the definition of diagnosis in each study. It is very difficult to read and figure out such information in current manuscript. - It is explained earlier in the text - that when the studies only enrolled BPS/IC patients the definition is the classic, in the article with hypersensitive bladder, the definition is the ESSIC criteria 1A which means that findings are normal and also the biopsies. The use of UCPPS term and patients is mainly to enrol also male patients. We agree that this brings some heterogeneity to the paper, but it is an intrinsic problem of this condition. However, we are the first to consider that a more robust and homogenous group of patients should be conducted multicentricly. 
  5. The site where VEGFs were evaluated varies among studies - urine & bladder. Please also provide the methodology of acquiring the specimen in each study. - Done.
  6. Overall, this study seems to be a review article rather than systematic review - We applied all the good practices and protocols for a systematic review.

Reviewer 2 Report

The introduction provided sufficient background and include all relevant references, the research design was appropriate, the methods were adequately described, the results were clearly presented and the conclusions were supported by the results.

Author Response

Thank you for your kind reply.

Best Regards